# Computational modeling of choice-induced preference change: A Reinforcement-Learning-based approach

Jianhong Zhu[1]*, Junya Hashimoto[2], Kentaro Katahira[3], Makoto Hirakawa[1], Takashi Nakao[1]

1 Graduate School of Humanities and Social Sciences, Hiroshima University, Hiroshima, Japan, 2 Graduate School of Education, Hiroshima University, Hiroshima, Japan, 3 Graduate School of Informatics, Nagoya University, Nagoya, Japan

* zhujianhong1995@gmail.com

**Data Availability Statement:** All original data files are available from the figshare database (DOI number 10.6084/m9.figshare.12083331).

## Abstract

The value learning process has been investigated using decision-making tasks with a correct answer specified by the external environment (externally guided decision-making, EDM). In EDM, people are required to adjust their choices based on feedback, and the learning process is generally explained by the reinforcement learning (RL) model. In addition to EDM, value is learned through internally guided decision-making (IDM), in which no correct answer defined by external circumstances is available, such as preference judgment. In IDM, it has been believed that the value of the chosen item is increased and that of the rejected item is decreased (choice-induced preference change; CIPC). An RL-based model called the choice-based learning (CBL) model had been proposed to describe CIPC, in which the values of chosen and/or rejected items are updated as if own choice were the correct answer. However, the validity of the CBL model has not been confirmed by fitting the model to IDM behavioral data. The present study aims to examine the CBL model in IDM. We conducted simulations, a preference judgment task for novel contour shapes, and applied computational model analyses to the behavioral data. The results showed that the CBL model with both the chosen and rejected value's updated were a good fit for the IDM behavioral data compared to the other candidate models. Although previous studies using subjective preference ratings had repeatedly reported changes only in one of the values of either the chosen or rejected items, we demonstrated for the first time both items' value changes were based solely on IDM choice behavioral data with computational model analyses.

## Introduction

### Externally guided decision-making (EDM) and computational modeling

The value learning process used by humans and animals has been investigated using a decision-making task with a correct answer specified by the external environment (externally

**Funding:** This research was supported by JST COI Grant Number JPMJCE1311(TN) and by JSPS KAKENHI Grants 18K03177(TN)and 18K03173 (KK). The funders had no role in study design, data collection and analysis, decision to publish, or preparation of the manuscript.

**Competing interests:** The authors have declared that no competing interests exist.

guided decision-making, EDM; [1]). In this case, people are required to adjust their choices based on feedback indicating a correct answer. The learning process of EDM is generally explained by the reinforcement learning (RL) model. In the typical RL model, the expected value (e.g., 0.8 is the expected value in the case of 1 dollar being rewarded with a probability of 80%) associated with the choice guides the choice behavior, and the expected value is updated in accordance with prediction error (i.e., the difference between the expected value and actually delivered reward) [2–4]. The appropriateness of the computational model is verified by fitting the model to trial-by-trial behavior choice data. Through the model-based analysis, latent variables can be estimated, such as the degree of value update (learning rate) as a model parameter, expected value, and prediction error of each trial. These estimated latent variables have been applied to neuroscience to understand the neural basis of EDM including social interaction [5–7].

## Internally guided decision-making (IDM) and computational modeling

In addition to the RL in EDM, the value of an item is learned through internally guided decision making (IDM) [1,8–20], in which no correct answer defined by external circumstances is available and one has to decide based on one's own internal criteria, such as preferences [1]. In IDM, it has been believed that the value of a chosen item is increased and that of a rejected item is decreased, called choice-induced preference change (CIPC; [8]). The IDM and EDM are distinguished conceptually, operationally, and from the neural bases [1,21–24]. Nevertheless, a choice-based learning (CBL) model [18,21,23,25,26] based on RL in EDM has been proposed for the IDM value-learning process. Thus the basic learning principles are assumed to have a commonality between EDM and IDM [18,22,25].

In the CBL model the values of both the chosen and rejected items are updated as if own choice were the correct answer. The CBL model was first proposed by Akaishi et al. [25] and revealed the tendency of people to make the same decision on perceptually ambiguous stimuli without feedback. In their model, the choice itself serves as feedback and the value of the choice estimate is updated as if the previous choice was the correct choice. Although Akaishi et al. [25] used a perceptual decision-making task, which was more externally guided than internally guided [1], they proposed that the same mechanism can be applied to the CIPC in IDM. Based on Akaishi's model, Nakao et al. [22,23] constructed the CBL model for IDM and conducted simulations to show that the behavioral index they used (i.e., change of decision consistency) is observed as the result of CBL.

However, as Nakao et al. [22,23] described, they could not test the appropriateness of the CBL model in IDM by fitting the model to the actual behavioral data. The stimuli used in their IDM task (i.e., occupation words for an occupation preference judgment task) are not appropriate for the application of computational model analysis. The stimuli are subject to participants' initial preferences formed through their daily life experiences before the experiment, and the existence of differences in initial value among stimuli makes it difficult to estimate model parameters properly, such as learning rate [27]. No studies have overcome this methodological limitation to apply computation analysis to IDM behavioral data.

The first aim of the present study was to examine the appropriateness of the CBL model in IDM by fitting the model to actual behavioral data. We applied the following two strategies to overcome the methodological limitation to applying computation analysis to IDM behavioral data. First, we used preference judgment for novel contour shapes as an IDM task. In EDM studies, novel contour shapes have been used, the initial value of which can be assumed to be equal, and applied to computational model analysis (e.g., [28–30]). By following the EDM studies, we used novel contour shapes to minimize the distortion of model parameter

estimation caused by the different initial preferences. Second, we compared the CBL model with a control model without free parameters (e.g., learning rate) in which the probability of choosing an option was estimated based on the chosen frequency in the previous trials. The control model shared the assumption that there was no difference in initial preference among novel contour shapes in the CBL model and preference was estimated based on choice history. Hence, even if the effect of an initial value difference could not be completely ruled out by using novel contour shapes, a better fit of the CBL model to behavioral data than the control model indicated the CBL model's appropriateness for incorporating free parameter.

## Are values of both chosen and rejected items changed in IDM?

As is common in CIPC studies using model-free classical CIPC paradigms, only the value change in either the chosen or rejected item or no change has been frequently reported (e.g., [14,15,18,20]), whereas it is widely believed that the values of both the chosen and rejected items change in CIPC (e.g., [15,18,20]). Although the reason for the unilateral value change and/or the small effect size of CIPC [19] is unclear, the measurement of the value change (CIPC) has been based on subjective preference ratings before and after the preference decision making, and that might lead to the non-robust experimental results. Indeed, the subjective rating is contaminated with rating noise [19] and is not always consistent with the choice behavior [31]. These pieces of evidence suggest the importance of establishing a method for examining CIPC without using subjective ratings. Besides, not using a subjective rating is also useful to avoid pseudo CIPC [12,19], which is caused in cases where both of the following procedures are applied: (1) noise-contaminated subjective ratings to measure CIPC, and (2) calculating the preference changes separately for chosen and rejected items. Although rate-rate choice [12,14,16] and blind choice [13,15] paradigms using subjective rating have been developed to avoid pseudo CIPC, the effect size is small [19]. Developing a method that does not use noise-contaminated subjective rating would lead to a more robust observation of CIPC.

The second aim was to investigate whether the value of both chosen and rejected items changed without using subjective ratings. To achieve this aim, we compared the CBL model with both values changed and with the models with either the value of chosen or rejected item changed. Note that although we collected subjective preference ratings for all novel contour shapes after the IDM (i.e., preference choice) task, the rating data were not for the computational model analysis to examine the CIPC, but to examine the inconsistency between subjective ratings and choice behavior suggested by the previous study [31].

## Improvements in CBL-based models

In EDM studies, the RL models incorporating various parameters, such as reducing decision noise with the accumulation of experience in decision-making [32] and value forgetting over time [33], have been proposed. With reference to those EDM studies, as the third aim of the present study, we explored the possible modification of the CBL model to explain CIPC better.

## Aims

The general aim of the present study was to examine the value learning process in IDM (i.e., CIPC) by applying an RL-based CBL model to behavioral data. More specifically, as we have described above, we examined (1) the appropriateness of the CBL model in IDM, (2) whether the value of both chosen and rejected items changed, and (3) the possible modification of the CBL model. We addressed the first and second aims in Study 1 and the third aim in Study 2.

In both studies, first, we conducted simulations for parameter and model recoveries to confirm whether the models acceptably identified the actual parameter and model. We generated

500 sets of artificial choice data using each model with the same setting with the experiment, and fit the model to the artificial data. For parameter recovery, we examined whether the parameters used for generating the artificial data were successfully estimated by the model fitting in each model, while for model recovery, we examined whether the models used to generate the artificial data show the best fit to their own artificial data. We then conducted a behavioral experiment and used the computational models to analyze actual choice behavioral data.

## Study 1

In Study 1, we examined (1) the CBL model's appropriateness by using preference judgment with novel contour shapes and a control model, and (2) whether the value of both chosen and rejected items changed or not. The control model for the first aim estimated the probability of selecting an option based on the chosen frequency in the previous trials without free parameters. The control model shared with the CBL models the assumption that there is no difference in the initial value among the novel contour shapes. For the second aim, we compared the four types of CBL models describing how participants updated items' values in the series of preference judgment tasks (Table 1). The four CBL models differed in whether only the value of chosen or rejected items changed (CBL$\alpha_c$ and CBL$\alpha_r$, respectively), or whether both values changed with the same or different learning rates (CBL$\alpha_{cr}$ and CBL$\alpha_c\alpha_r$ respectively).

Regarding the first aim, we expected CBL models to fit better than the control model with the series of choice behaviors in IDM. Concerning the second aim, we expected the CBL model with both learning rates for chosen and rejected item (CBL$\alpha_{cr}$ and/or CBL$\alpha_c\alpha_r$) to fit better than the other CBL models with either of the learning rates (CBL$\alpha_c$ and CBL$\alpha_r$).

## Method

### Participants

Forty-eight healthy Japanese university students (male = 21, female = 27, mean age = 21, age range = 18–36) participated in the behavioral experiment. All participants were native Japanese speakers, right-handed, with normal or corrected-to-normal vision. The study was approved by the ethics committee of the Graduate School of Education, Hiroshima University. According to the guidelines of the research ethics committee of Hiroshima University, all participants provided written informed consent. They were paid for their participation in the experiment.

### Stimuli and apparatus

From Endo et al. [34], we selected 15 novel contour shapes with moderate complexity, width, smoothness, symmetry, orientation and association values (Table 2). The IDs of the shapes

Table 1. Summary of the differences among the four computational models.

| CBL models | Degree of value change after decision | |
| --- | --- | --- |
| | chosen | rejected |
| CBL$\alpha_c$ | $\alpha_c(1-Q_i(t))$ | 0 |
| CBL$\alpha_r$ | 0 | $\alpha_r(1-Q_i(t))$ |
| CBL$\alpha_{cr}$ | $\alpha_{cr}(1-Q_i(t))$ | $\alpha_{cr}(1-Q_i(t))$ |
| CBL$\alpha_c\alpha_r$ | $\alpha_c(1-Q_i(t))$ | $\alpha_r(1-Q_i(t))$ |

Note. $\alpha$ denotes learning rate, and $Q_i(t)$ denotes value of item $i$ at trial $t$. $\alpha_c$ and $\alpha_r$ represent the different values of learning rates applied to the chosen and rejected items, while $\alpha_{cr}$ denotes that the learning rate of chosen and rejected items are the same value.

**Table 2. Summaries of the geometrical properties of 15 novel contour shapes.**

|  | Complexity | Width | Smoothness | Symmetry | Orientation | Association |
|---|---|---|---|---|---|---|
| **Mean** | 5.09 | 5.54 | 4.60 | 3.65 | 5.03 | 65.7 |
| **SD** | 1.42 | 1.81 | 1.67 | 1.72 | 2.14 | 6.84 |

were 29, 31, 35, 36, 37, 39, 42, 44, 45, 56, 63, 65, 81, 87, and 92. The size of the image used in the experiment is $800 \times 600$ pixels. Participants could see a picture on the screen within 30 degrees of the angle of view. All possible combinations of 15 shapes (i.e., 105 pairs) were generated.

In each trial, one of the 105 pairs was presented with one member on the left and the other on the right side of the screen on a white background via Psychopy [35]. The order of trials and presentation sides of shapes were randomized across participants. The display screen size was $1920 \times 1080$ and the experiment was run on a Windows 10 PC.

## Preference judgment task

Participants performed 5 blocks of 21 trials of a preference judgment task. Participants were instructed to choose the preferred one from the two presented shapes. There were 105 combinations of stimuli, and each stimulus pair was presented only once. Each stimulus was presented 14 times. Before the experimental trials, participants were given four practice trials to familiarize themselves with the experimental process. The stimuli used in the practice were different from those used in the experimental trials.

Each trial began with the fixation cross shown for a randomly selected period of 2,000 ms, 2,400ms, or 2,600ms (Fig 1). After that, two shapes were presented on the left and right side of the fixation cross for 2,000 ms. Participants indicated their choice with the "f" (left) or the "j" (right) key on a standard computer keyboard as quickly and accurately as possible after the shapes were presented. Although the two stimuli disappeared (i.e., turned to white screen) after 2,000 ms to control the exposure time for each stimulus, participants could make their decision by pressing the key after the stimulus disappeared. The white screen was presented until the participants' keypress for preference judgment. If the key was pressed within 2,000 ms, the stimuli were also displayed until the end of the display time (i.e., 2,000 ms) and the white screen was not presented. The reaction time (RT) from the presentation of the two stimuli to the response was recorded. After each block of 21 trials, the participants pressed any key to continue the task once they had rested enough.

In order to examine the consistency of preference judgment and subjective rating, the participants conducted the subjective preference rating task after the preference judgment task. In the rating task, participants were instructed to evaluate their subjective preference for each of the 15 novel contour shapes, rated on a 5-point scale (1 = *extremely dislike*, 5 = *extremely like*). The data of the preference ratings of the Likert scale were not used in the computational model analysis.

## CBL models

In the traditional reinforcement learning model, the expected value of the chosen option is learned through a series of previous behavior results and used to decide a later choice. The learning process of the reinforcement learning model can be written as

$$Q(t + 1) = Q(t) + \alpha(r(t) - Q(t)) \tag{1}$$

The combined strength $Q(0 \leqq Q \leqq 1)$ between the choice (conditional stimulus) and the

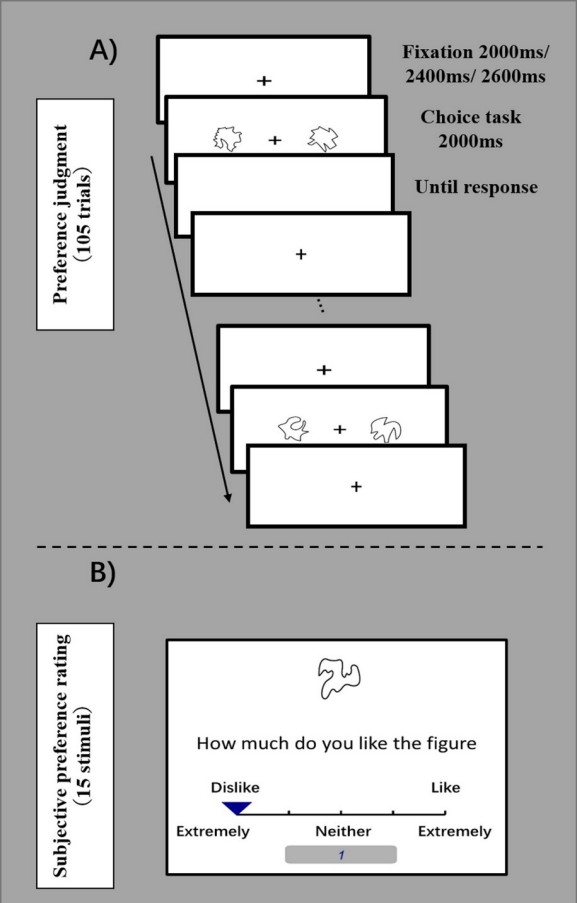

**Fig 1. Experimental procedure of IDM task and subjective rating.** A) In the IDM task, participants were asked to choose his/her preferred option from two stimuli. The stimuli display time was 2,000 ms. In case participants did not respond during the period, the screen displayed a blank screen until the response. B) In a subjective preference rating, participants subjectively evaluated all 15 stimuli on a 5-point scale (1 = *extremely dislike*, 5 = *extremely like*). The data of the subjective rating were not used for the computational model analysis.

reward (unconditional stimulus) is updated in each trial $t$. $r (0 \leqq r \leqq 1)$ represents the degree of compensation in the trial $t$. $\alpha (0 \leqq \alpha \leqq 1)$ is a parameter that determines the degree of updating of $V$ in one trial (learning rate).

Previous simulation studies [22,23] applied the Rescorla-Wagner reinforcement learning rule [36] to build CBL models. The learning process of the CBL models is that the value of chosen items increases while the value of rejected items decreases. The learning process of CBL model can be written:

$$V_i(t+X) = \begin{cases} V_i(t) + \alpha(1 - V_i(t)) \text{ if } i \text{ was chosen} \\ V_i(t) + \alpha(0 - V_i(t)) \text{ if } i \text{ was rejected} \end{cases} \qquad (2)$$

where the value $V_i(t)$ of option $i$ is updated at trial $t$ and the updated $V$ value is held unchanged until the item is presented after $X$ trials (c.f., F-CBL model in Study 2). The learning rate $\alpha$ and prediction error ($1 - V_i(t)$ or $0 - V_i(t)$, depending on whether it was chosen or rejected by a participant) determined the degree of value change followed by choice. The learning rate takes a value between 0 and 1.

The difference between the values of the two options was applied to the following softmax function to calculate the choice probability.

$$P_{chosen} = \frac{1}{1 + \exp(-\beta(V_{chosen}(t) - V_{rejected}(t))}$$ (3)

$P_{chosen}$ indicates the probability that the model selects the option actually chosen by a participant. $V_{chosen}$ and $V_{rejected}$ and indicates the estimated value of each chosen and rejected item. $\beta$ determines the slope of the softmax function. The higher $\beta$ is, the more the judgment is based on the values, while the lower $\beta$ is, the more random the choice is and more independent of the value.

We prepared four models based on the RL model (see Table 1). Since it is possible that only the chosen or rejected items' value changes occurred as in previous CIPC studies (e.g., [14,20]), we prepared models in which only the chosen or rejected items' value changes (CBL$\alpha_c$ and CBL$\alpha_r$, respectively). In addition, we also prepared two models in which the values of both the chosen and rejected items change, with either the same (CBL$\alpha_{cr}$) or different learning rates (CBL$\alpha_c\alpha_r$).

**Control model without free parameters.** To examine the validity of setting the free parameters (i.e., $\alpha$ and $\beta$) in the CBL models, we conducted behavioral data analyses using a control condition without free parameters. In the control model, the value $V$ of the chosen item $i$ is given by

$$V_i(t) = \frac{N_i^{chosen}(1:t-1)}{N_i^{presented}(1:t-1)}$$ (4)

$N_i^{presented}(1:t-1)$ is the number of times item $i$ is presented before trial $t$. $N_i^{chosen}(1:t-1)$ is the number of times item $i$ is chosen among them. $V_i$ is 0 when the item $i$ is presented for the first time.

The probability that the model selects the option chosen by a participant ($P_{chosen}$) is given by

$$P_{chosen} = \frac{V_{chosen}(t) + 1}{V_{chosen}(t) + V_{rejected}(t) + 2}$$ (5)

To define the probability, even if both the chosen and rejected items' values are zero, one and two were added to the numerator and the denominator, respectively (see, Ito and Doya [37]).

## Simulations and model-based behavioral data analyses

The following two simulations and model-fits to the actual behavioral data were run on Matlab (www.mathworks.com). We used Matlab's fmincon function to estimate the free parameters, such as the learning rate ($\alpha$) and the gradient of the softmax function ($\beta$) that maximizes each CBL model's likelihood of generating behavioral data.

**Simulation 1 (Parameter recovery).** First, we conducted Simulation 1 (parameter recovery) to confirm whether the experimental design and each model satisfy the goal of estimating the model parameters from the choice data [38]. More specifically, in each model, we examined whether the model parameters used to generate artificial behavioral data were successfully estimated after fitting the model to the artificial behavioral data.

In each of the models, simulations were conducted 100 times. When generating the artificial behavioral dataset, we used the same settings as the actual experimental design. That is, the number of stimuli was 15 and the trial consisted of the possible combinations of these (i.e., 105

trials). In this study, we used novel contour shapes to assume the initial values of stimuli are consistent when applying computational model analysis. We, therefore, set the initial value for all stimuli to 0.5 in the CBL models. The initial value of all items in the control model was set to 0 since the number of times all items presented was 0. The range of parameters set for each model when generating artificial behavioral datasets is shown in Table 3. "$U$" indicates a uniform distribution, and the value of $\alpha$ was randomly selected from a uniform distribution on the interval (0, 1), and the value of $\beta$ was randomly selected from a uniform distribution on the interval (0, 20). We set $\beta$ in that range because a larger value of $\beta$ generates random choice behavioral data, which reflects the differences of the models less.

We estimated the best fitting parameters for each artificial behavioral dataset by the maximum likelihood method. The log-likelihood (*LL*) for all the trials was written in terms of the choice probabilities of the individual model for the chosen option in the behavioral data as

$$LL = \sum\nolimits_{t=1}^{T} \log P_{chosen}(t) \tag{6}$$

where $T$ represents the total number of trials. The $P_{chosen}$ is the probability that the model selects the stimuli actually chosen by a participant in each trial $t$, calculated based on formulas (2) and (3). We used Matlab's fmincon function with an interior-point method [39] to estimate the free parameters that maximize the likelihood of each model generating behavioral data. Optimality tolerance and step tolerance were set to 1e-8. The search range of $\alpha$ and $\beta$ were the same as the range set during artificial data generation.

**Simulation 2 (model recovery).**   We conducted Simulation 2 (model recovery) to confirm whether the true model showed the best fit to the behavioral data generated from that model. To evaluate the relative goodness of fit of the models, the AIC (Akaike Information Criterion) was used, which is given by

$$AIC = -2LL + 2k \tag{7}$$

where $k$ is the number of free parameters. A smaller AIC denotes a better model fit to the data.

We simulated the behavior of the four CBL and control models in the preference judgment. As in Simulation 1, artificial behavioral data were generated. The method of model parameter estimation was also the same as in Simulation 1. Each generated dataset was then fit to each of the given models to determine which model showed the best fit (according to AIC). Then 500 artificial datasets per each model were generated, and the frequency of being the best fit model was calculated for each fitted model.

**CBL model fit to the behavioral data.**   We conducted computational model analyses of the actual behavioral data. The method of model parameter estimation was the same as Simulations 1 and 2. However, to deal with the possibility that the actual value of $\beta$ could be greater than 20, we performed an additional analysis with the search range of $\beta$ set to 0–100. The AICs were calculated using all the models that fulfilled the criteria in the two simulations by applying the choice data of each participant to the model. The AICs were tested using the multiple-comparison Holm method, adjusted for all possible combinations of the model comparisons.

Table 3. Parameter settings for each model (Simulations 1 and 2).

| Models | Parameters |
|---|---|
| CBL$\alpha_c$ | $\alpha_c \sim U(0,1), \beta \sim U(0,20)$ |
| CBL$\alpha_r$ | $\alpha_r \sim U(0,1), \beta \sim U(0,20)$ |
| CBL$\alpha_{cr}$ | $\alpha_{cr} \sim U(0,1), \beta \sim U(0,20)$ |
| CBL$\alpha_c\alpha_r$ | $\alpha_c \sim U(0,1), \alpha_r \sim U(0,1), \beta \sim U(0,20)$ |

For an intuitive understanding of how much the models predicted the behavioral data, we calculated the normalized likelihood [37] given by

$$z_L = e^{(LL/T)} \tag{8}$$

where $LL$ is the log-likelihood calculated based on formula (6), and $T$ represents the total number of trials. The normalized likelihood represents the averaged probability per trial of the model selecting the item actually chosen by a participant. This index presents 0.5 when the probability of the model choosing the actually chosen item by a participant is a chance-level. A larger normalized likelihood denotes better model prediction to the behavioral data. However, different from the AIC, the normalized likelihood was not adjusted for the impact of the number of free parameters.

### RT data analysis

To confirm whether the participants faithfully conducted preference judgment, we compared RTs between the large (choice between two similarly preferred items) and small conflict (choice between preferred and not preferred items) conditions. Previous studies reported that preference judgment takes longer for large than small conflict trials [21,40–42]. We divided trials into large and small conflict trials on the preference judgment by calculating the difference between the chosen frequencies of two stimuli in each trial. More specifically, we first calculated the chosen frequency across trials for each stimulus, and then calculated the difference in the chosen frequency between the two stimuli in each trial. Trials with differences in chosen frequency less than the average (i.e., the preferences for the two stimuli were similar) were assigned to the large conflict condition, while those with differences in chosen frequency greater than the average were assigned to the small conflict condition. Finally, for each participant, we calculated the mean RT in each conflict condition. The RTs outside the mean ± 3$SD$ within each participant were excluded from the analyses.

Besides, we conducted a correlation analysis between RTs and the difference between the chosen frequencies of the two stimuli in each trial. The trials with a larger difference in the chosen frequency corresponded to those with a smaller difference. In each participant, the data of the mean ± 3$SD$ were excluded, and the Pearson's correlation coefficient ($r$) between RT and conflict was calculated. The $r$ values were converted to Fisher's $Z$ to conduct a one-sample $t$-test against 0 (i.e., no correlation).

**Rating data analysis.** To examine the consistency between the preference judgment and subjective rating, we first counted the frequency of each participant's choice for each stimulus and applied a median split to divide the stimuli into high frequency stimuli and low frequency stimuli. The average subjective rating scores for these two types of stimuli were compared.

We also conducted a correlation analysis between the chosen frequency and subjective rating of the stimuli, as well as a reaction time data analysis.

## Results

### Results of Simulation 1 (Parameter recovery)

In Simulation 1, we confirmed that the parameters of the computational model during the generation of artificial data could be properly calculated by applying the same computational model to the artificial data. We found good consistency between the set parameter values (simulated) and estimated values (fit) (Fig 2, $r$s > .77), confirming that the parameters of the models could be calculated when the initial values of the stimuli were equal.

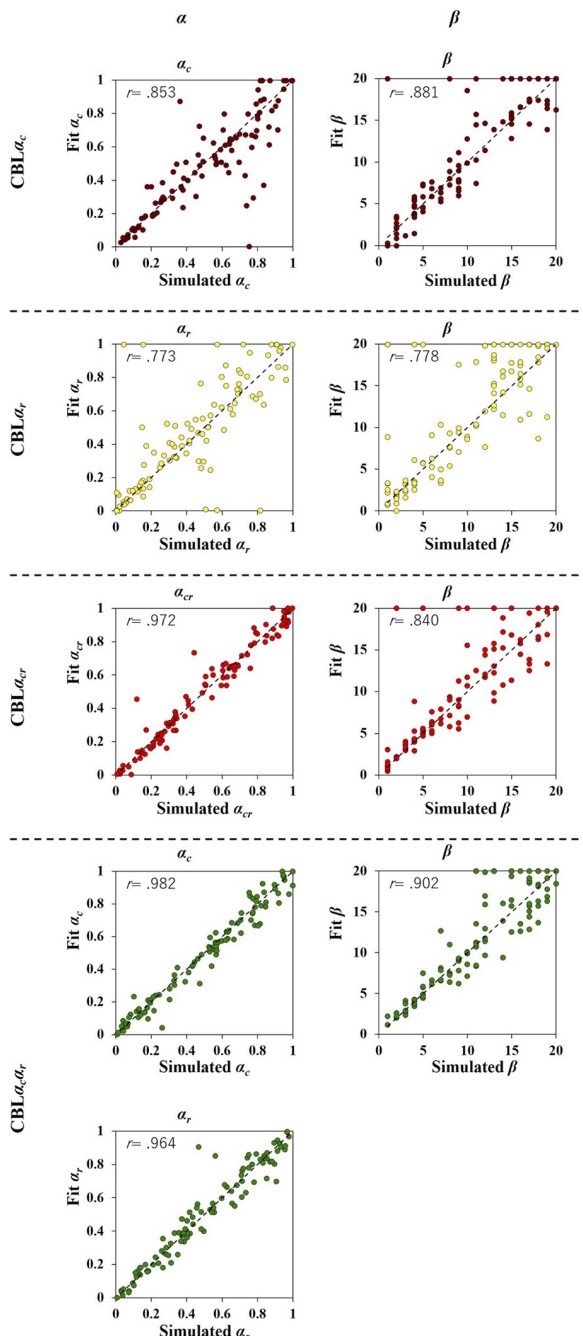

**Fig 2. Results of parameter recovery in Study 1 (Simulation 1).** A parameter recovery was conducted to confirm whether each model satisfied estimating the model parameters from the choice data. The correlation coefficients between the parameters used to generate the artificial data are shown as parameter recovery indices. All model parameters used to generate artificial behavioral data were successfully estimated by fitting the same models.

## Results of Simulation 2 (Model recovery)

Fig 3 shows the mixed matrix resulting from the model recovery. When each model was used to generate artificial data and the respective model showed better fit to the data it generated than the other models, that model can be regarded as able to recover the true model from the data. When

**Simulated model**

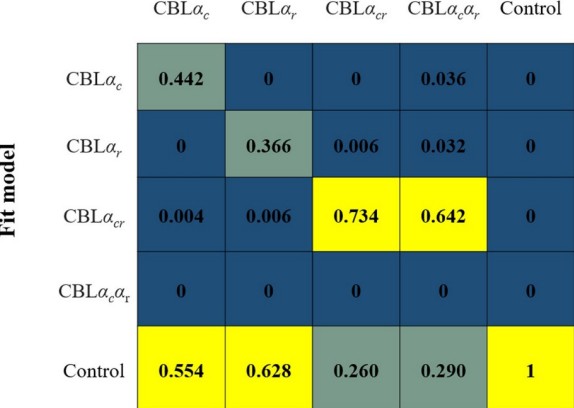

**Fig 3. Results of model recovery in Study 1 (Simulation 2).** A model recovery was conducted to confirm whether the true model showed the best-fit to the behavioral data generated from that model. The artificial data generated by each simulated model were used for model fit. The model with the smallest AIC was adopted as the best fit model for the data. Each model generated 500 sets of artificial data and the data were fitted to all models. The value in each cell indicates the proportion of the best fit model in the 500 sets of artificial data in each simulated model.

the control model and CBL$\alpha_{cr}$ models were the true models, the model could be recovered by the respective model. However, when CBL$\alpha_c$ or CBL$\alpha_r$ were the true models, the control model showed the best fit. This meant that we could not determine whether CBL$\alpha_c$, CBL$\alpha_r$, or the control model was the true model when the control model showed the best fit. Besides, when CBL$\alpha_c\alpha_r$ was the true model, CBL$\alpha_{cr}$ showed the best fit. Thus, we could not distinguish whether the true model was CBL$\alpha_{cr}$ or CBL$\alpha_c\alpha_r$ when CBL$\alpha_{cr}$ showed the best fit. In other words, from the present study, we thus could not clarify whether the learning rates between chosen and rejected items differed.

One plausible reason for the failure of the CBL$\alpha_c\alpha_r$ model recovery is that in the cases the randomly selected $\alpha_c$ and $\alpha_r$ to generate the artificial data using CBL$\alpha_c\alpha_r$ were similar to each other, CBL$\alpha_{cr}$ fitted the artificial data. Meanwhile, CBL$\alpha_c$ or CBL$\alpha_r$ fitted best when there was a large difference in the learning rates. In fact, as we can see in Fig 3, compared with the cases where the CBL$\alpha_{cr}$ model generated the artificial data, there are more cases where CBL$\alpha_c$ or CBL$\alpha_r$ provided the best fit when CBL$\alpha_c\alpha_r$ generated the artificial data. Besides, as shown in Table 4, the difference between $\alpha_c$ and $\alpha_r$ to generate the artificial data by CBL$\alpha_c\alpha_r$ was larger in the cases the CBL$\alpha_c$ or CBL$\alpha_r$ adopted than the case of the CBL$\alpha_{cr}$ was the best.

## Results of CBL and control model fit to the experimental behavioral data

Fig 4 shows a comparison between the models where the search range for alpha and beta were 0–1 and 0–20, respectively. All the CBL models showed a better fit to the behavioral data than the control model ($ts(47) > 11.30$, $ps < .001$; Holm). CBL$\alpha_{cr}$ showed the best fit for the

**Table 4. Summary of randomly determined learning rates to generate artificial data using CBLαcαr in the cases where the other three CBL models showed the best fit (Simulations 2).**

| The best fit model | Proportion of artificial data where $\alpha_c > \alpha_r$ | Mean $\alpha_c$ | Mean $\alpha_r$ |
|---|---|---|---|
| CBL$\alpha_c$ | 100% | 0.58 | 0.07 |
| CBL$\alpha_r$ | 0% | 0.03 | 0.60 |
| CBL$\alpha_{cr}$ | 50% | 0.52 | 0.53 |

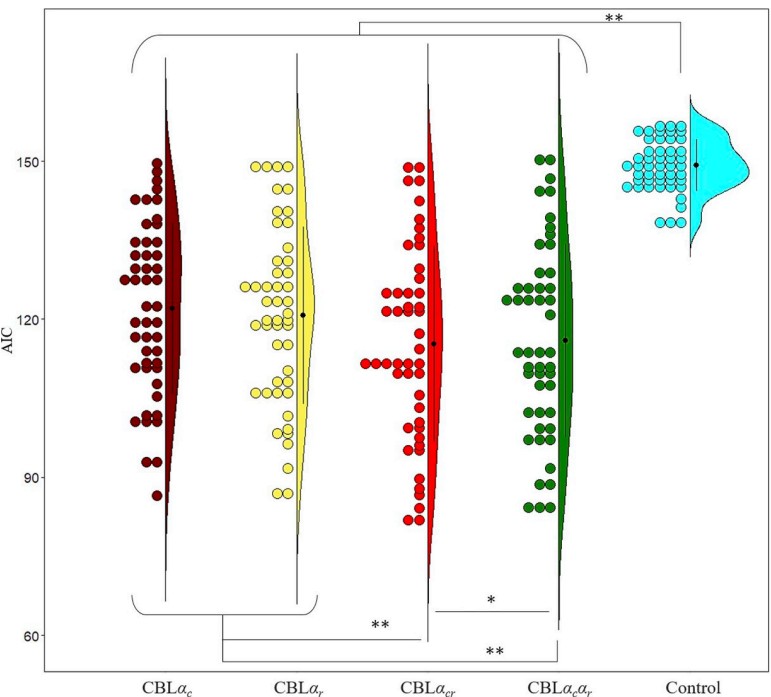

**Fig 4. AIC results after fitting each model to actual behavioral data (Study 1).** * $p < .05$, ** $p < .001$. The black dots, error bars, and colored dots indicate the average, standard deviation (*SD*), and each participant's data, respectively.

behavior data ($ts(47) > 2.47$, $ps < .035$; Holm). When we compared AIC at an individual participant level, 16.67%, 10.42%, 60.41%, 10.42%, and 2.08% participants showed the best fit to the CBL$\alpha_c$, CBL$\alpha_r$, CBL$\alpha_{cr}$, CBL$\alpha_c\alpha_r$, and control models, respectively.

The normalized likelihood for CBL$\alpha_c$, CBL$\alpha_r$, CBL$\alpha_{cr}$, CBL$\alpha_c\alpha_r$, and the control models were 57.17% (*SE* = 0.64), 57.55% (*SE* = 0.67), 59.08% (*SE* = 0.75), 59.47% (*SE* = 0.74), and 49.16% (*SE* = 0.16), respectively. Similar to the results of the AIC, all the CBL models showed higher probability to select the actually chosen option than the control model ($ts(47) > 12.21$, $ps < .001$; Holm). CBL$\alpha_{cr}$ showed a significantly higher probability than the other models ($ts(47) > 6.37$, $ps < .001$; Holm), except CBL$\alpha_c\alpha_r$. CBL$\alpha_c\alpha_r$ showed a significantly higher probability than CBL$\alpha_{cr}$ ($t(47) = 5.39$, $p < .001$; Holm).

In cases where the search range of $\beta$ was set to 0–100, similar results were observed to when the search range was 0–20, except that no significance was found between CBL$\alpha_{cr}$ and CBL$\alpha_c\alpha_r$. The mean AICs for CBL$\alpha_c$, CBL$\alpha_r$, CBL$\alpha_{cr}$, CBL$\alpha_c\alpha_r$, and the control models were 121.49 (*SE* = 2.40), 120.28 (*SE* = 2.48), 115.14 (*SE* = 2.71), 115.71 (*SE* = 2.69), and 149.16 (*SE* = 0.71), respectively. All the CBL models showed better fits to the behavioral data than the control model ($ts(47) > 11.23$, $ps < .001$; Holm). CBL$\alpha_{cr}$ ($ts(47) > 6.14$, $ps < .001$; Holm) and CBL$\alpha_c\alpha_r$ ($ts(47) > 5.98$, $ps < .001$; Holm) showed better fits than the other models.

Taken together, the CBL models showed better fits than the control model. Besides, the CBL$\alpha_{cr}$ and CBL$\alpha c\alpha r$ models in which both the updated chosen and rejected values showed better fits than CBL$\alpha_c$ and CBL$\alpha_r$ in which only one of the value were updated.

## Results of RT

We examined participants' RT in the large and small conflict conditions of preference judgment. We found that participants' reactions took longer in the large than the small conflict

condition ($t(47)$ = 3.68, $p < .001$; Fig 5A). This result was consistent with those of previous studies [21,40–42]. Besides, we observed significant correlation between RT and conflict (mean $Z$ = -0.21, $SE$ = 0.02, $t(47)$ = -8.77, $p < .001$). The scatter plot between the RT and conflict from all the trials of all the participants is shown in Fig 5B.

These results were consistent with previous studies [21,23,42], and indicated that participants had conducted the preference judgment task faithfully.

## Results of subjective rating

We compared the subjective preference ratings of frequently chosen stimuli (high-frequency; HF) and less frequently chosen stimuli (low-frequency; LF). No significant difference was found between the two conditions ($t(47)$ = −.313, $p > .75$, Fig 6A). Besides, no significant correlation was found between chosen frequency and subjective rating (mean $Z$ = 0.02, $SE$ = 0.04, $t(47)$ = 0.40, $p$ = 0.69; Fig 6B). Thus, the preference reflected in choice behavior was not reflected in subjective rating after the preference judgment task, which is consistent with the results of the research of Katahira et al. [31].

## Discussion

Study 1 aimed to examine (1) the CBL model's appropriateness by comparing CBL models with the control model, and (2) whether the value of both chosen and rejected items changed or not.

Regarding the first aim, CBL models showed a better fit to IDM behavioral data than the control model (Fig 4). Although the results of Simulation 2 (Fig 3) showed that the control model was the best fit for a certain proportion of data even when the CBL (especially, CBL$\alpha_c$ and CBL$\alpha_r$) models were the true model, the CBL model did not have the best fit when the control model was the true model. Since the CBL models had a better fit than the control model to the actual behavioral data, it was unlikely that the control model was the true model. The control and CBL models similarly assumed no difference in initial preference among novel contour shapes and estimated value based on the preference choice history. Since the main difference was the presence of free parameters (i.e., learning rate and reverse temperature), the better fit of the CBL models to behavioral data than the control model confirmed the CBL model's appropriateness for incorporating free parameters.

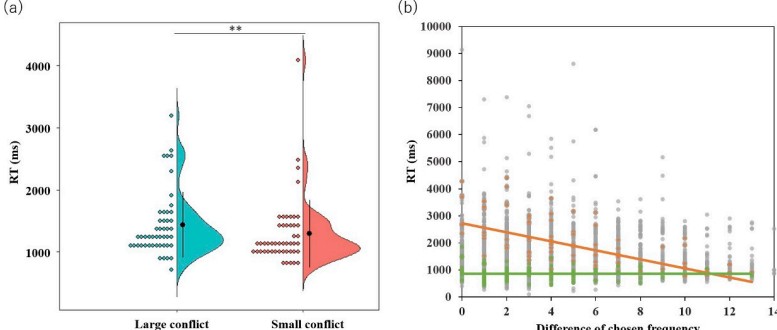

**Fig 5. The relationships between the degree of conflict and reaction time (RT).** (a) The mean average of RT in high- and low-conflict conditions (back dots), $SD$, and each participant's data (colored dots). ** $p < .001$. (b) A scatter plot of all the trial data of all the participants between RT and the difference of chosen frequency. The trials with a larger difference in the chosen frequency corresponded to those with less conflict. The black regression line corresponds with all the data points of all the participants. The orange and green dots and the regression lines correspond with the participants' data that showed the strongest negative correlation and the data that did not, respectively.

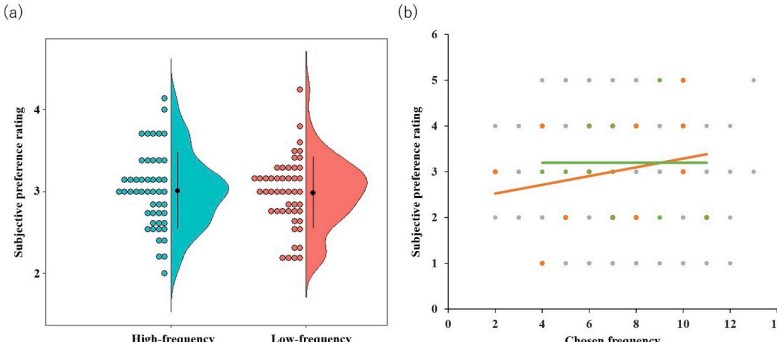

**Fig 6. The relationship between subjective preference rating and chosen frequency in a preference judgment task.**
(a) Mean subjective preference ratings for frequently chosen (high-frequency) and less chosen (low-frequency) stimuli
(back dots), *SD* (error bars), and each participant's data (colored dots). (b) A scatter plot of all stimulus data from all
participants between subjective preference rating and chosen frequency. The black regression line corresponds to all
data points of all participants. The orange and green dots and regression lines corresponding to participant's data
showed the strongest negative correlation and those that did not, respectively.

With regard to the second aim, although the CBL$\alpha_c\alpha_r$ result of the model fit to the experimental behavioral data was unreliable because the model failed model recovery in Simulation 2, CBL$\alpha_{cr}$ did fit the behavioral data better than CBL$\alpha_c$ and CBL$\alpha_r$. This result verified that the values of chosen and rejected items are updated in IDM, as had been believed in the CIPC studies. Although previous CIPC studies using subjective preference ratings had repeatedly reported changes only in one of the values of either chosen or rejected items, we demonstrated for the first time both items' value changes based solely on IDM choice behavioral data with computational model analyses.

Importantly, although CBL$\alpha_{cr}$ was adopted in model comparisons, there remained the possibility that the CBL$\alpha_c\alpha_r$ was a true model and that was likely useful for parameter estimation of individual learning. Although CBL$\alpha_c\alpha_r$ had failed model recovery in Simulation 2, as can be seen in Fig 3, the results of this model recovery indicated that the true model could be CBL$\alpha_c\alpha_r$ when CBL$\alpha_{cr}$ was the best fit. Besides, in Simulation 1, the CBL$\alpha_c\alpha_r$ model showed good parameter recovery. Furthermore, although averaged AIC by fitting the actual participants' data was better in CBL$\alpha_{cr}$ than in CBL$\alpha_c$, CBL$\alpha_r$, and CBL$\alpha_c\alpha_r$ (Fig 4), when we compared AIC in an individual participant's data, 16.67%, 10.42%, and 10.42% of participants showed a best fit to CBL$\alpha_c$, CBL$\alpha_r$ and CBL$\alpha_c\alpha_r$, respectively. For those participants' data, the estimating parameters in CBL$\alpha_{cr}$ were likely to be inappropriate. For parameter estimation in individual participants, including those who had different learning rates for chosen and rejected items, it was possible that applying CBL$\alpha_c\alpha_r$, which is a model that encompasses the CBL$\alpha_c$ and CBL$\alpha_r$, that CBL$\alpha_{cr}$ would be plausible.

The present study's subjective ratings were not consistent with choice behavior (Fig 6). Subjective ratings were contaminated by rating noise [19]; it is known that there is a discrepancy between the values reflected in the subjective rating and those actually reflected in the choice behavior [31]. This could be the reason for the low effect size [19] in the paradigm of assessing CIPC from a combination of subjective ratings and behavioral choices. That was one of the reasons why we introduced the CBL model analysis to examine CIPC without using subjective ratings. However, it should be noted that the stimuli in the present study were presented 14 times and selected multiple times unlike the typical CIPC experiment [8]. Such multiple selections could lead to large discrepancies between choice behavior (i.e., the chosen frequency in preference judgment) and subjective rating. Therefore, it is important to note that the discrepancy between choice behavioral data and subjective ratings in this study does not negate the paradigm's usefulness using subjective ratings.

## Study 2

In Study 2, we did an exploratory investigation into the possible modification of the CBL model to explain CIPC better. We constructed and validate modified Tβ-CBL (CBLɑ$_{cr}$ model with time-varying β) and F-CBL (CBLɑ$_{cr}$ model with the forgetting parameter for unpresented items) based on the CBLɑ$_{cr}$ model that had a best fit to the behavioral data in Study 1.

The Tβ-CBL was constructed to examine the possibility that the more a stimulus was experienced, the less noise was included in the IDM. To investigate this possibility, we set the β as a variable over time by following Yechiam et al. [32].

The F-CBL model was developed to explore the possibility that the value of the items not presented decayed over time. In the CBL models in Study 1, the values of the chosen and the rejected stimuli updated, while the value of the stimulus that was not present in a trial remained unchanged. To examine the possibility of attenuations of the values of unpresented stimuli, we added the forgetting factor ($\alpha_F$) to the CBLɑ$_{cr}$ model.

The examination of the F-CBL model corresponded to the examination of trial-based temporal autocorrelation in IDM. The CIPC was considered to cause autocorrelation in IDM: chosen items' preference increased and items were subsequently more likely to be chosen, while rejected items' preference decreased and items were subsequently less likely to be chosen. The CBL model was one of the computational models that represented such autocorrelation. The CIPC was reflected in the time series behavioral data as a stimulus-based autocorrelation because two out of 15 stimuli were presented in the present study task and that could be captured by CBLɑ$_{cr}$ model. However, it was also possible that the item chosen in the recent past was more likely to be chosen, that is, a trial-based temporal autocorrelation. Since the F-CBL included attenuation of the value of stimuli not presented in each trial, this model would be the best fit if there were a trial-based temporal autocorrelation in the IDM.

## Methods

### Modified CBL models

*Tβ-CBL model*: In this model, β increases with increasing experiences (i.e., the number of times the items are presented).

$$\beta = \left(\frac{N_L^{presented}(1:t) + N_R^{presented}(1:t)}{2} \cdot \frac{1}{10}\right)^{\wedge c} \tag{9}$$

$N_L^{presented}(1:t)$ and $N_R^{presented}(1:t)$ is the number of times the stimuli shown at the left and right sides on the screen are presented until the trial $t$. $c$ is a free parameter that modulates the degree of increase in β with experience. The higher the $c$ value, the clearer the preference judgment as experience the presentation and judgment of the stimuli.

*F-CBL model*: In this model, the value $V$ of the items $i$ that did not present in each trial $t$ was updated as follows:

$$V_i(t+1) = (1 - \alpha_F) \times V_i(t) \text{ if } i \text{ was not presented} \tag{10}$$

where $\alpha_F$ is the forgetting factor which modulates the degree of attenuation of the value. The forgetting factor was included by following the previous EDM study [33]. In this model, the forgetting factor was applied to unchosen items, unlike the present study.

### Simulations and model-based behavioral data analyses

The simulations for parameter and model recovery were conducted in the same way as Study 1. Parameter recovery in Simulation 3 was conducted for the Tβ-CBL and F-CBL models.

Model recovery in Simulation 4 included the CBL$\alpha_{cr}$, T$\beta$-CBL, and F-CBL models. In both the simulations, the range of $c$ in the T$\beta$-CBL model was set to (0, 9), corresponding approximately to the range of $\beta$ (0, 20) in the CBL$\alpha_{cr}$ model. The range of the forgetting factor ($\alpha_F$) in the F-CBL model was set to (0, 1), the same as the learning rate by following the previous study, which included the forgetting factor for the RL model [33].

## Results

### Simulation 3 (Parameter recovery)

Overall, different to Simulation 1, the correlations between the simulated and fitted parameters were not strong in the both the T$\beta$-CBL and F-CBL models (Fig 7, $r$s > .40).

### Results of Simulation 4 (Model recovery)

Fig 8 shows the mixed matrix resulting from the model recovery. When the T$\beta$-CBL model was the true model, the CBL$\alpha_{cr}$ model also fitted best in the same proportion as T$\beta$-CBL model. The other two models were able to recover a true model.

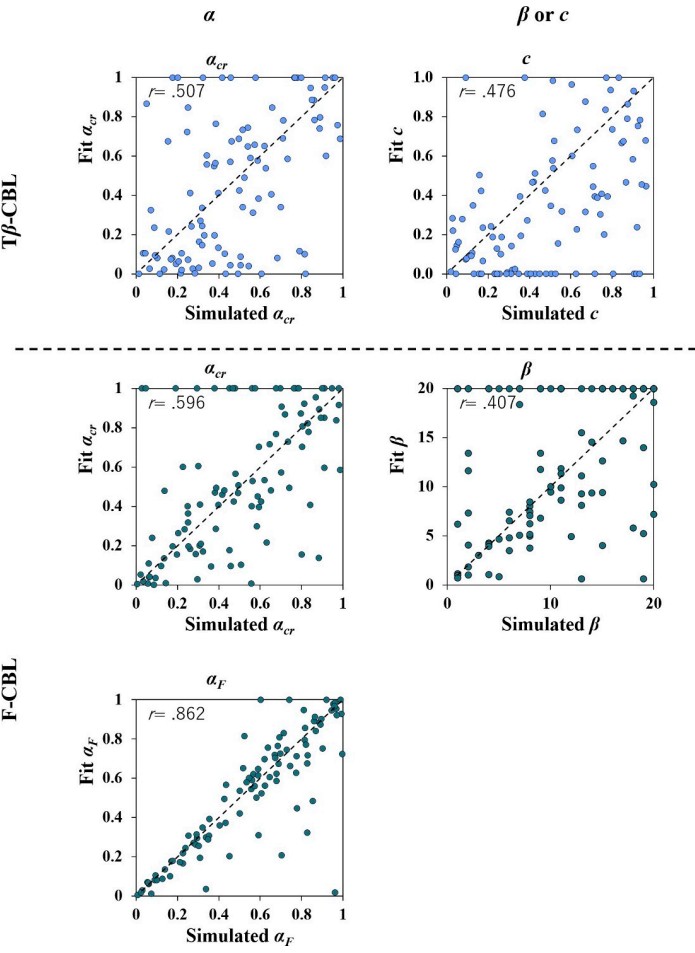

**Fig 7. Results of parameter recovery in Study 2 (Simulation 3).** The correlation coefficient between the parameter used to generate artificial data is shown as indices of parameter recovery.

## Simulated model

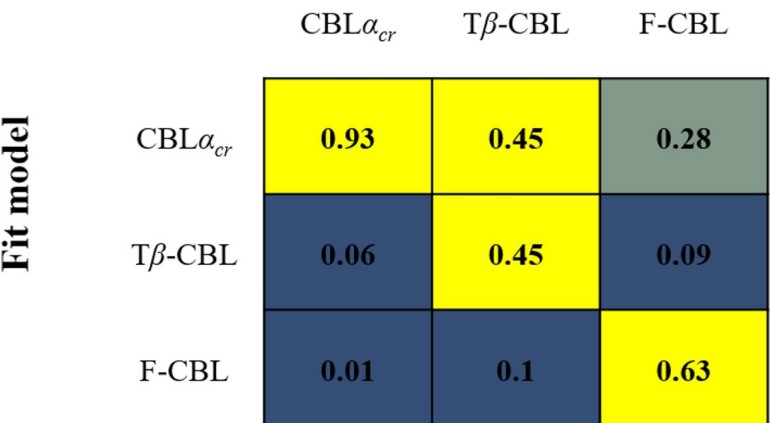

**Fig 8. Results of model recovery in Study 2 (Simulation 4).** The model with the smallest AIC was adopted as the best-fit model to the data. Each model generated 500 sets of artificial data and the data were fitted to all the models. The value in each cell indicates the proportion of the best-fit model in 500 sets of artificial data in each simulated model.

### Results of model fit to the experimental behavioral data

Fig 9 shows a comparison between the models. The result for CBL$\alpha_{cr}$ is the same as in Fig 4 in Study 1. CBL$\alpha_{cr}$ model showed better fits to the behavioral data than the other two models ($ts$ (47) > 7.55, $ps$ < .001; Holm). CBL$\alpha_{cr}$ showed the best fit for behavior data ($ts$(47) > 2.47, $ps$ < .035; Holm). When we compared AIC on an individual participant level, 83.33%, 20.08%, and 14.58% of the participants showed a best fit to the CBL$\alpha_{cr}$, T$\beta$-CBL, and F-CBL models, respectively.

The normalized likelihoods for CBL$\alpha_{cr}$, T$\beta$-CBL, and F-CBL were 59.08% ($SE$ = 0.75), 54.04% ($SE$ = 0.26), and 59.20% ($SE$ = 0.75), respectively. CBL$\alpha_{cr}$ and F-CBL showed higher probability to select the actually chosen option than T$\beta$-CBL ($ts$(47) > 9.34, $ps$ < .001; Holm). F-CBL showed a higher probability than CBL$\alpha_{cr}$ ($t$(47) = 2.27, $p$ = .028; Holm).

### Discussion

Study 2 was an exploratory investigation into the possible modification of the CBL model. In the comparison of AIC, the CBL$\alpha_{cr}$ model showed a better fit to IDM behavioral data than the T$\beta$-CBL and F-CBL models (Fig 9). In contrast, F-CBL showed a better fit in normalized likelihood than CBL$\alpha_{cr}$ and there was a discrepancy between normalized likelihood and the AIC results. This discrepancy was likely caused by no adjustment for the number of free parameters in normalized likelihood. The F-CBL model had one more free parameter than the CBL$\alpha_{cr}$ model. The results of Simulation 4 (Fig 8) suggested that if the F-CBL model were the true model, there would be a similar number of participants for whom the CBL$\alpha_{cr}$ and F-CBL models would be the best fit. In the actual behavioral data results, however, 83.33% of the participants showed a best fit on the CBLacr model. Thus, it could be concluded that the CBL$\alpha_{cr}$ was a better fit than the other two models in the present study.

In the first place, however, it should be noted that the parameter recovery of the T$\beta$-CBL and F-CBL models was not as good as that of the CBL$\alpha_{cr}$. That is, it remains possible that the

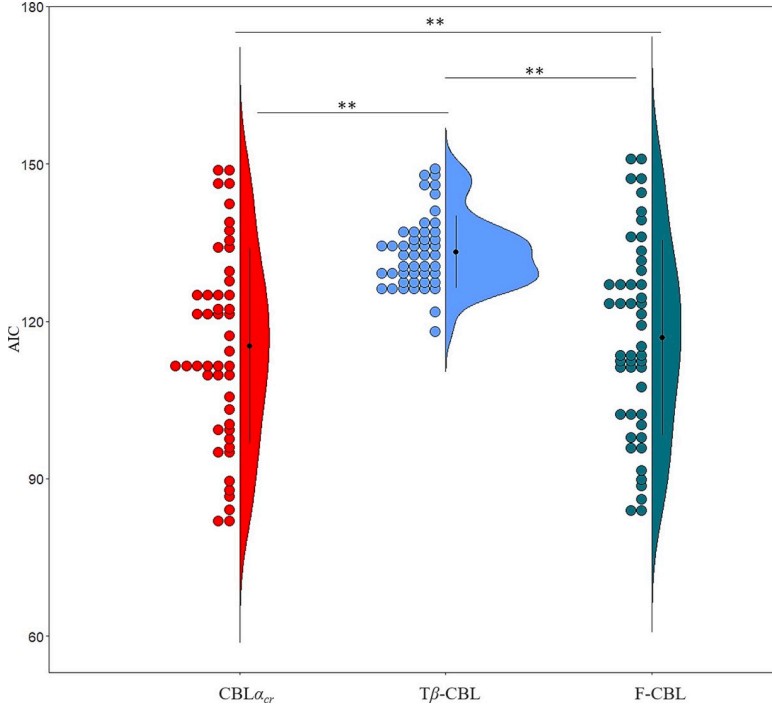

**Fig 9. AIC results after fitting each model to actual behavioral data (Study 2).** $^{**}$ $p < .001$. The black dots, error bars, and colored dots indicate the average, *SD*, and each participant's data, respectively.

experimental paradigm of the present study was not designed well enough to validate the two modified CBL models. For example, looking at the results of parameter recovery for the F-CBL model (Fig 7), which was prepared to examine trial-based temporal autocorrelation, adding the forgetting factor $\alpha_F$ in the CBL$\alpha_{cr}$ model reduced the accuracy of $\alpha_{cr}$ estimation (c.f., Fig 2). The learning rate $\alpha_{cr}$ led to a stimulus-based autocorrelation such that the chosen item was more likely to be chosen when presented later. Theoretically, in the present paradigm, the same item was presented every seven trials on average. Hence, it is possible that stimulus-based autocorrelation and trial-based autocorrelation are not easily estimated separately. Future experiments with an increased number of items may be useful to evaluate the F-CBL model properly.

## General discussion

The general aim of this study was to examine the value learning process in IDM (i.e., CIPC) by applying the RL-based CBL model to behavioral data. The first aim of the present study was to examine the appropriateness of the CBL model in IDM by fitting the model to actual behavioral data. In Study 1, by using novel contour shapes and comparing with the control model without free parameters (e.g., learning rate), we showed the appropriateness of the CBL models to describe CIPC in IDM. The second aim was to investigate whether both chosen and rejected items' value changed without using subjective ratings. In Study 1, we also compared four CBL models (CBL$\alpha_c$, CBL$\alpha_r$, CBL$\alpha_{cr}$, and CBL$\alpha_c\alpha_r$). Although previous CIPC studies using subjective preference ratings had repeatedly reported changes only in one of the values of either chosen or rejected items, we demonstrated for the first time both items' value changes during IDM, based solely on choice behavioral data with computational model analyses as EDM studies.

The third aim was to explore the possible modification of the CBL model. From the computational model analysis of the behavioral data in Study 2, the CBL$\alpha_{cr}$ model fitted better than the other modified models, T$\beta$-CBL and F-CBL models. However, the parameter recovery of the two modified models was not as good as the CBL$\alpha_{cr}$ model and there remained the possibility that the present study's paradigm was not appropriate for considering these modifications.

Taken together, it was shown that the CBL$\alpha_{cr}$ model (and also potentially CBL$\alpha_c\alpha_r$, see Discussion in Study 1) could explain CIPC in IDM, the potential for further model improvement remains, though.

## Cognitive dissonance theory

The phenomenon of CIPC has been explained by the theory of cognitive dissonance [43], according to which choosing one item from two items with the same preference subjective rating produces dissonance feelings, which is called "cognitive dissonance." After that, people adjust their preferences to support their choice as reasonable. That is, they increase their preference for the chosen items and decrease their preference for the rejected items to restore cognitive consistency. Although the present study's model analysis was not limited to trials with the two items with similar values (i.e., preference), the results supported CIPC. However, by incorporating dissonance into the CBL$\alpha_{cr}$ model it could be a better model to explain CIPC in IDM, and this possibility should be explored in a future study with an appropriate experimental paradigm for that.

In the context of studies of cognitive dissonance theory in CIPC, it has been debated when the preference change occurs, during the choice phase or during the post-choice subjective rating phase. Lee and Daunizeau [26] indicated that the process of restoring cognitive consistency occurs during the decision process rather than during the post-ratings. In addition, Nakao et al. [22] reported a link between CIPC and brain activity around 400 ms after choice. The present study also provides evidence that the CIPC occurs during decision choice since our model-based analysis involved applied choice behavior data.

Vinckier et al. [44] used a computational model to examine the cognitive dissonance theory. However, they used an effort task in which participants required effort to get a reward (i.e., foods) and experienced success or failure, and examined the preference change for the foods. The task was closer to EDM than IDM. Although the types of decision-making they addressed were different to those of the present study, it was obvious that preference change did not occur only through simple IDM. Besides, daily decision-making was not simple IDM as focused on in this study, but involved complex factors such as effort, which had been treated in EDM. Future research could consider integrating our model with the models considered by Vinckier et al. [44] and others for an integrated understanding of the human decision-making process.

## Limitations and further directions

Although our study confirmed the validity of CBL model in IDM, the following limitations must be considered. First, the present study showed that the CBL model could describe CIPC by assuming the values of chosen and rejected items are updated as if own choice were the correct answer. However, we did not compare the CBL model with cognitive dissonance theory. To clarify the best model to illustrate CIPC, it would be necessary to compare the CBL model with a computational model that reflected the impact of cognitive dissonance in a future study. Second, the present study could not test the possibility that the learning rates of the chosen and rejected items differed by comparing models by fitting to behavioral data. Since some

participants had a best fit with Models 1 and 2, it was suggested that Model 4 might be appropriate for an individual's parameter estimation. However, it would be necessary to explore an experimental paradigm that can demonstrate the validity of using Model 4 in a model comparison. Third, as we discussed in Study 2, to examine the validity of the modified CBL model (i.e., TB-CBL and F-CBL), it is desirable to conduct experiments with appropriate task design to test those models.

## Conclusion

In this study, we carried out the preference judgment task with novel contour shapes to apply computational modeling to IDM. From simulations and a behavioral experiment with computational modeling as with EDM, we showed that the CBL model could describe CIPC, and we confirmed that the value of both the chosen and rejected items change without using subjective ratings. The computational modeling allowed us to estimate trial-by-trial preference change and model parameters (e.g., learning rate). Those parameters could apply for further analyses, such as for neural bases of IDM, as had been done in EDM research. That would lead to an increase in an integrated understanding of EDM and IDM; for example, whether learning rates, which have common psychological meanings for EDM and IDM, are common in the neural substrates.

## Acknowledgments

Please refer to the following link for the raw data: Zhu, Jianhong; Hashimoto, Junya; Katahira, Kentaro; Hirakawa, Makoto; Nakao, Takashi (2020): Computational Modeling of Choice-Induced Preference Change: A Reinforcement-Learning-based approach. figshare dataset. https://doi.org/10.6084/m9.figshare.12083331.v1.

## Author Contributions

**Writing – original draft:** Jianhong Zhu.

**Writing – review & editing:** Junya Hashimoto, Kentaro Katahira, Makoto Hirakawa, Takashi Nakao.

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
