## [Decision Letter · Decision Letter 0]

11 Jun 2020

PONE-D-20-10026

Computational modeling of Choice-Induced Preference Change: A Reinforcement-Learning-based approach

PLOS ONE

Dear Dr. Zhu,

Thank you for submitting your manuscript to PLOS ONE. After careful consideration, we feel that it has merit but does not fully meet PLOS ONE’s publication criteria as it currently stands. Therefore, we invite you to submit a revised version of the manuscript that addresses the points raised during the review process.

We recommend that it should be revised taking into account the changes requested by the reviewers. Since the requested changes includes Major Revision, the revised manuscript will undergo the next round of review by the same reviewers.

We look forward to receiving your revised manuscript.

Kind regards,

Baogui Xin, Ph.D.

Academic Editor

PLOS ONE

Journal Requirements:

Reviewers' comments:

Reviewer's Responses to Questions

**Comments to the Author**

1. Is the manuscript technically sound, and do the data support the conclusions?

Reviewer #1: Yes

Reviewer #2: Yes

Reviewer #3: Partly

Reviewer #4: Partly

2. Has the statistical analysis been performed appropriately and rigorously? 

Reviewer #1: I Don't Know

Reviewer #2: Yes

Reviewer #3: No

Reviewer #4: Yes

3. Have the authors made all data underlying the findings in their manuscript fully available?

Reviewer #1: Yes

Reviewer #2: Yes

Reviewer #3: No

Reviewer #4: Yes

4. Is the manuscript presented in an intelligible fashion and written in standard English?

Reviewer #1: Yes

Reviewer #2: Yes

Reviewer #3: Yes

Reviewer #4: Yes

5. Review Comments to the Author

Reviewer #1: General: Zhu et al., present results from a study of decision-making, in which they analyze internally guided decision-making within a reinforcement learning context called choice-based learning (CBL) to describe a phenomenon of internal decision making called choice-induced preference change (CPIC). At a high level, the theory of estimating the perceived rewards (Q value) starting from an initial arbitrary decision towards a goal of maintaining internal consistency is interesting, and if validated could provide an important consideration in analysis of decision-making. The specific simulation approach and parameter estimation function (fmincon) are outside of my area of expertise, and I’ll have to defer to experts in this area for specific commentary on the internal validity of the simulations performed. My main concern regarding the approach is that it is not clear to what extent the authors definitively ruled out prior external experience (value estimation) as biasing the initial choices. Other comments are given below.

Major:

-- Although the authors provide a solid explanation for how a CBL model for CPIC can be simulated using novel contour shapes, it is not clear to me how rigorous this approach is for avoiding differential initial preferences. In other words, how do we know that a subject does not have an initial preference for a given shape before the experiment? Is there a way to test this alternative possibility, or are we left to just assume based on intuition?

-- Figure 1 indicates that preference to images is provided using a Likert scale, although the computational models use Softmax function (actually it looks more like a logistic function) to calculate probabilities, and based on the description provided, choices are dichotomized. Was the ordinal relationship within the Likert scale addressed, and if not, why was this scale used in the experiment rather than a binary decision?

-- Did the authors address autocorrelation within an individual’s selections? It would seem that the time between preference choices would be correlated, although it is not clear whether any of the models accounts for temporal autocorrelation, as for example, a hidden Markov model might.

-- Since PLOS One is aimed toward more general readership, the authors might consider widening the scope with more explanation about the applications of this approach within the field of neurology/psychology and behavior.

Minor:

-- Perhaps it reflects my distance from the subject matter, but the initial statement ‘the value of an item is learned through making a series of decisions,’ (line 45) seems fairly abstract and no further information is given to support what context that value is made? What ‘item’ is being referred to? A decision tool? An item of value being purchased? As above, some additional context in the Background could be helpful.

-- In most RL applications, the learning rate is a hyperparameter, and thus is assigned outside the modeling process. On line 56, the authors note that the learning rate is estimated as a model parameter. If there is another method the authors are applying to identify a learning rate, it should be provided, otherwise this section should be corrected to indicate that LR is a hyperparameter, and must be optimized through manual or automated search (e.g., grid search), not fit like a statistical parameter. If this optimization is all performed by fmincon function, then more information about the specific algorithm is needed for us less informed readers.

-- Table 2 is referenced before Table 1

-- How was the range of B of [0, 20] chosen?

Reviewer #2: The paper presents validity of the CBL model that has not been previously confirmed by fitting the model to IDM behavioral data, since the differences of initial preferences among the items make it difficult to estimate the model parameters. Hence, the paper presents a set of experiments conducted with novel contour shapes that are supposed to be equaly probably selected.

The paper is very well writen and explain clearly the main concepts. In addition, the discussion part has very clear and concise arguments that support the experiments results.

Figure 2 should be created with more definition as it is blurred when the size is big.

Reviewer #3: In this study, the authors suggested the validity of the choice-based learning model in internally guided decision-making (IDM).

Please see attached word file for my full review on this paper.

Many Thanks.

Reviewer #4: I am not sure if most researchers in the reward and RL worlds would accept the formulation of IDM. Is it really necessary to build this work on this notion? What if we do not learn the value through a series of decisions, but simply your valuations are noise estimates and by experience we sharpen them? In that case the reward is still external, but we just have to learn the value of the external reward. That seems more general to me, because 0.8 of 1 dollar also has a utility to me personally I need to learn, I know the expected value (0.8) but the utility I have to learn by experience.

In fact I find the setup unexpected. So if I understood this correctly, the introduction claims that this does not work with the usual stimuli that have intrinsic values (like pictures or jobs) that are classically used for the rate-choice-rate task? But instead, this task uses arbitrary shapes. And now the logic is that the choice is the supervision signal to adjust the evaluation? So in a way, this is promoting internal consistency?

In the classic way of studying this, the rate-choose-rate procedure is accompanied by a rate-rate-choose control condition that estimates the baseline amount of increased consistency. How is this model accounting for that effect in the baseline condition? Does it need this control condition too? Presumably the parameter estimates should be smaller in this condition because there cannot be a causal link? (chen and risen 2010 and many papers that cite it)

If I understand it correctly in the current study the conditions are just ’choose-rate’ because it is assumed that all shapes are very close in value. It seems to me a rate-choose control condition is necessary?

I was surprised that this paper is not discussed ‘Sour grapes and sweet victories: How actions shape preferences’, how the two modelling approaches are related appears to be relevant?

The results in figure 6 are very surprising are they not? The way I understand it, subject chose between items 14 times, but their final appraisals (the ‘rate’ part of the classic rate-choose-rate) shows no trace of this? Doesn't this mean there is no choice induced chance in the valuation, and it is purely an increased consistency in the choices?

Is there a way of looking at this with higher resolution, by plotting the subjects rating for each of the 15 items to a. The number of times the item was chosen b. The value that the Rl model assigned to that stimulus? Would this results mean that in a classic rate-choose-rate setup, we would conclude that ‘it didn't work’ because there is no effect of choice on rate?

The discussion mentions two limitations, I am most concerned about the former.It is not really clear to me how to interpret these results and what conclusions to draw from this model, particularly in contrast to existing models. My confusion is increased by the results in figure 6.

Also can you add a URL where the data is in the paper, i couldn't actually make the figshare link work

I hope my comments are helpful to make this a bit clearer, it is a nice idea!

6. PLOS authors have the option to publish the peer review history of their article (what does this mean?). If published, this will include your full peer review and any attached files.

Reviewer #1: No

Reviewer #2: No

Reviewer #3: Yes: Takashi Nakano

Reviewer #4: No

---

## [Author Response · Author response to Decision Letter 0]

9 Dec 2020

We are grateful for the excellent and extremely helpful comments. We addressed all the various issues, and we hope that the manuscript has now been improved considerably. Please let us know if further changes are necessary. We are more than happy to carry out such changes.

For responses to specific reviewers' comments, see "Response_To_reviewers".

Modifications made to the main text and are shown using yellow highlights (see the separate version labeled “Manuscript with changes marked”).

---

## [Editor Report · Decision Letter 1]

10 Dec 2020

Computational modeling of Choice-Induced Preference Change: A Reinforcement-Learning-based approach

PONE-D-20-10026R1

Dear Dr. Zhu,

We’re pleased to inform you that your manuscript has been judged scientifically suitable for publication and will be formally accepted for publication once it meets all outstanding technical requirements.

Kind regards,

Baogui Xin, Ph.D.

Academic Editor

PLOS ONE
---

## [Editor Report · Acceptance letter]

14 Dec 2020

PONE-D-20-10026R1 

Computational modeling of Choice-Induced Preference Change: A Reinforcement-Learning-based approach 

Dear Dr. Zhu:

I'm pleased to inform you that your manuscript has been deemed suitable for publication in PLOS ONE. Congratulations! Your manuscript is now with our production department. 

Kind regards, 

on behalf of

Professor Baogui Xin 

Academic Editor

PLOS ONE